# Chemical evolution of an autonomous DNAzyme with allele-specific gene silencing activity

Kim Nguyen [1,5], Turnee N. Malik[1,5] & John C. Chaput [1,2,3,4] ✉

Low activity has been the primary obstacle impeding the use of DNA enzymes (DNAzymes) as gene silencing agents in clinical applications. Here we describe the chemical evolution of a DNAzyme with strong catalytic activity under near physiological conditions. The enzyme achieves ~65 turnovers in 30 minutes, a feat only previously witnessed by the unmodified parent sequence under forcing conditions of elevated $Mg^{2+}$ and pH. Structural constraints imposed by the chemical modifications drive catalysis toward a highly preferred UG̲UD motif (cut site underlined) that was validated by positive and negative predictions. Biochemical assays support an autonomous RNA cleavage mechanism independent of RNase H1 engagement. Consistent with its strong catalytic activity, the enzyme exhibits persistent allele-specific knock-down of an endogenous mRNA encoding an undruggable oncogenic KRAS target. Together, these results demonstrate that chemical evolution offers a powerful approach for discovering new chemotype combinations that can imbue DNAzymes with the physicochemical properties necessary to support therapeutic applications.

RNA-cleaving DNA enzymes (DNAzymes or Dz), such as Dz 10–23, offer an attractive modality for targeting undruggable regions of the human genome[1,2]. These reagents benefit from inexpensive and scalable protocols for manufacturing designer sequences by solid-phase chemical synthesis, programmable binding arms that can be engineered to recognize nearly any RNA target, a highly specific cleavage mechanism that dramatically reduces activity against non-homologous sequences, and high safety and tolerability margins in human patients[3,4]. However, despite their many advantages, interest in DNAzymes as gene silencing agents in therapeutic applications has declined due to their low efficacy in clinical trials. This problem, which is primarily due to their poor catalytic activity under physiological conditions, raises an outstanding challenge in the field of nucleic acid chemistry of how to develop DNAzymes that can maintain high catalytic activity in cellular environments where the concentration of free magnesium ($Mg^{2+}$) is limiting[5,6].

The last 30 years have witnessed tremendous growth in the evolution of chemical modifications for oligonucleotide therapeutics. Although antisense oligonucleotides (ASOs) and small interfering RNAs (siRNA) were the primary focus of this effort[7,8], chemical modifications have also improved the stability and catalytic activity of DNAzymes[9]. Modifications commonly observed in DNAzyme scaffolds include the addition of inverted 3′–3′ thymidine nucleotides[10], the substitution of phosphodiester linkages for phosphorothioate[11] or phosphoramidite linkages[12], and the replacement of natural DNA residues with unnatural xeno-nucleic acid (XNA) residues[13], such as 2′-*O*-methylribonucleic acid (OMe)[14], locked nucleic acid (LNA)[15], and 2′-fluoroarabinonucleic acid (FANA)[16]. Knowledge gained from these studies led to the development of new DNAzyme designs that function with improved activity, including a recent X10–23 design that exhibits substantial (>60%) reductions of mRNA and protein levels in cells[17].

[1]Department of Pharmaceutical Sciences, University of California, Irvine, CA 92697-3958, USA. [2]Department of Chemistry, University of California, Irvine, CA 92697-3958, USA. [3]Department of Molecular Biology and Biochemistry, University of California, Irvine, CA 92697-3958, USA. [4]Department of Chemical and Biomolecular Engineering, University of California, Irvine, CA 92697-3958, USA. [5]These authors contributed equally: Kim Nguyen, Turnee N. Malik. ✉ e-mail: jchaput@uci.edu

Elsewhere in the field, polymerase engineering has enabled the evolution of RNA-cleaving XNA enzymes (XNAzymes) composed entirely of non-natural nucleotides[18–21]. While these studies have narrowed the gap between DNAzymes and protein-based gene silencing agents, the ability to routinely generate DNAzymes with robust multiple turnover activity under physiological conditions remains an unsolved problem.

Here we report the chemical evolution of a highly modified 10−23 DNAzyme that functions with strong catalytic activity under near-physiological conditions. The enzyme was discovered through iterative cycles of design that were guided by structural information available on the folding topology and metal-ion binding sites of Dz 10−23[22]. The enzyme, termed Dz 46 (Fig. 1), can achieve ~65 turnovers in 30 min, making it a highly effective RNA-cleaving DNAzyme under near-physiological conditions. Structural constraints imposed by the chemical modifications drive catalysis toward a highly preferred UGUD motif (cut site underlined, and D is the IUBMB nucleotide code for A, G, or U) that was validated across several mRNA targets[23]. Detailed biochemical assays, including quantification of cellular RNase H1 levels, support an autonomous RNA cleavage mechanism with minimal RNase

H1 engagement. Consistent with its strong catalytic activity, the enzyme exhibits persistent allele-specific gene silencing activity in cultured mammalian cells, endogenously expressing a currently undruggable G12V mutant KRAS target. Our results demonstrate that chemical evolution offers a powerful approach to discovering new chemotype combinations that can be used to engineer DNAzymes with physicochemical properties that will help drive future clinical applications.

## Results

### Chemical evolution of Dz 10−23

Recent structural insights into the folding and dynamics of Dz 10−23 (Fig. 1A, Supplementary Fig. 1) provide a valuable starting point for designing next-generation DNAzymes that can function with improved stability and catalytic activity[22]. In particular, topological changes caused by metal−ion-induced conformational plasticity illuminate the importance of $dG_{14}$ as a critical residue in the catalytic loop. Molecular dynamics (MD) simulations indicate that the O2′ atom of the scissile $rG^0$ residue on the RNA substrate interacts with the catalytic $dG_{14}$

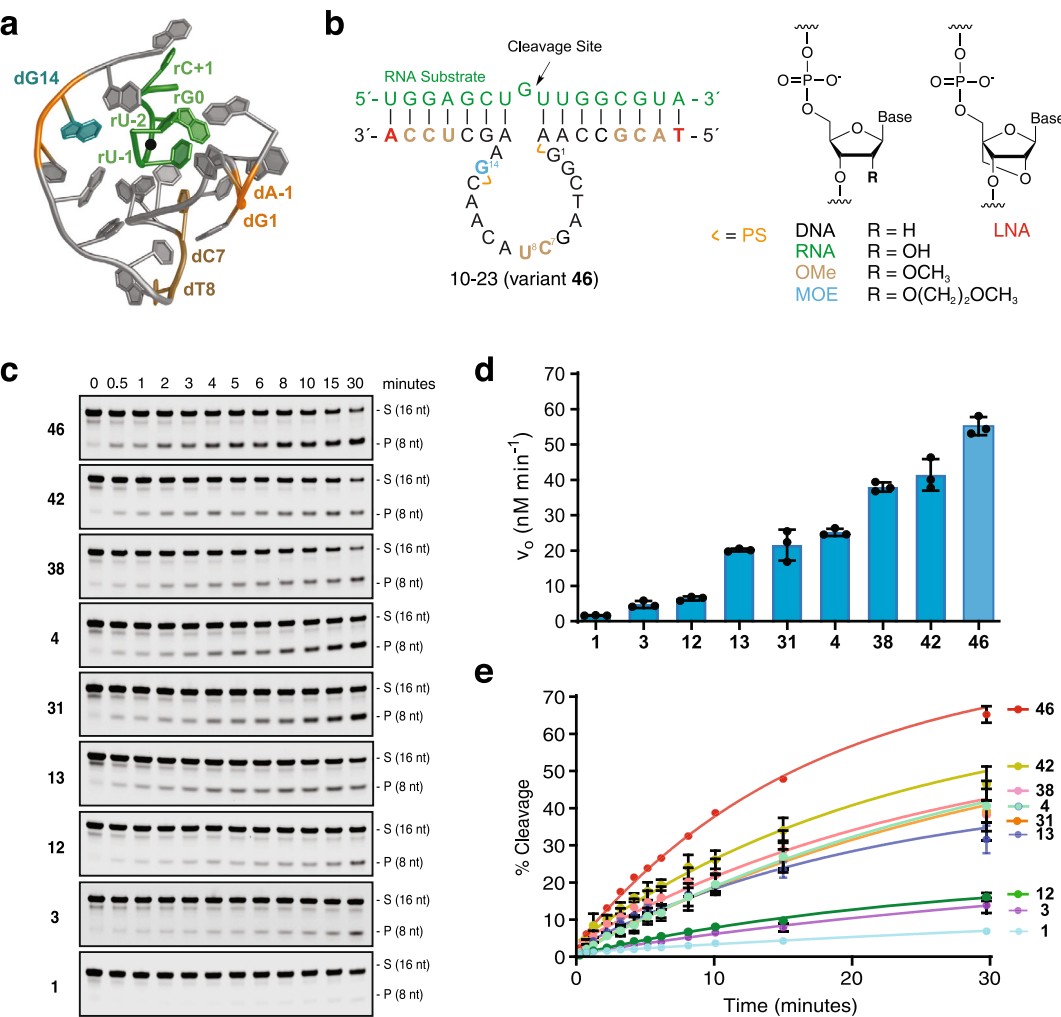

**Fig. 1 | Chemical evolution of DNAzyme 10-23. a** Cartoon representation of an NMR-averaged precatalytic structure of 10−23 bound to an RNA substrate (green) encoding a prion protein ("7PDU"). Dz is shown in gray, with colors and numbers matching residues found to be important in the chemically optimized version. **b** Left: Dz 10-23_v46 in complex with a 16-nt KRAS G12V RNA substrate. Right: Chemical structures of natural and modified nucleotides found in Dz 10-23_v46. Abbreviations: DNA (2′-deoxyribose nucleic acid), RNA (ribose nucleic acid), OMe (2′-deoxy-2′-methoxyribonucleic acid), MOE (2′-deoxy-2′-methoxyethoxyribonucleic acid), LNA (locked nucleic acid) and PS (phosphorothioate).

Modifications are colored to denote their position in Dz 10-23. **c** Representative denaturing PAGE gels showing time-dependent multiple turnover RNA cleavage activity across a panel of engineered Dz 10−23 variants. S: 5′-Cy5-labeled full-length substrate, P: 5′-Cy5-labeled cleavage product. **d, e** Initial velocities ($v_o$, D) and kinetic curves (E) were measured for each variant. Error bars denote the ±standard deviation of the mean for three independent replicates. All reactions were performed under simulated physiological conditions in a buffer containing 1 mM $MgCl_2$, 50 mM Tris (pH 7.5), 10 mM NaCl, and 140 mM KCl at 37 °C with 1000 nM substrate and 10 nM enzyme (100:1, S:E). Source data are provided as a Source Data file.

residue 5-times more frequently than its direct interactions with a hydrated $Mg^{2+}$ ion present in metal-ion-binding site III[22]. This observation suggests that 10–23 functions by an acid-base mechanism, whereby $dG_{14}$ abstracts a proton from the O2′ atom of the scissile $rG^0$ residue, while a hydrated $Mg^{2+}$ ion simultaneously donates a proton to the O5′ atom of the adjacent $U^{-1}$ residue. An in-line attack of the O2′ oxyanion on the adjacent phosphodiester bond leads to the formation of an upstream cleavage product carrying a cyclic 2′,3′-monophosphate, and a downstream strand with a 5′ hydroxyl group (Supplementary Fig. 2). Recognizing that the in-line conformation responsible for the proposed acid-base mechanism was observed with low frequency in MD simulations[22], we postulated that synthetic congeners of DNA introduced at position $dG_{14}$ and elsewhere in the scaffold could lead to improved catalytic activity by shifting the equilibrium of the various 10–23 topologies in favor of a catalytically active conformation.

We envisioned a chemical evolution process in which the classic Dz 10–23 framework would be optimized through a series of iterative design, build, test, and learn cycles intended to uncover permissive sites in the scaffold where chemical modifications could contribute favorably to catalytic activity. We focused our designs around an 8 + 7 binding arm configuration, which provided a balanced solution to the problem of how to enhance enzyme kinetics while avoiding product inhibition caused by increased thermodynamics of substrate binding. We recognized that certain chemical modifications, especially those made to the bind arm region, could alter the equilibrium between the pre- and post-catalytic state and therefore sought to identify synthetic congeners that would maximize multiple turnover activity through optimal RNA binding. To ensure that the best designs functioned in a cellular context, all of the optimization assays were performed under multiple turnover conditions in buffer meant to simulate the cellular environment [1 mM $MgCl_2$, 140 mM KCl, 10 mM NaCl, and 50 mM Tris-HCl (pH 7.5)] at 37 °C. The functional impact of each chemical perturbation toward Dz 10–23 activity was measured by denaturing polyacrylamide gel electrophoresis (PAGE) (Supplementary Figs. 3–10, Supplementary Tables 1–4). As the chemical complexity of our variants increased, the substrate (S) to enzyme (E) ratio was increased from 10:1 to 100:1 (S:E) so that we could accurately distinguish the activity of our best variants, some of which function on the timescale of seconds under the less stringent conditions of the 10-fold excess substrate (Supplementary Figs. 11 and 12).

Influenced by the NMR structure of Dz 10–23, we began by evaluating the RNA cleavage activity of 10-23 variants carrying synthetic congeners at position $dG_{14}$ of the catalytic loop. Substitution of the 2′-deoxyguanosine nucleotide for a panel of base-matched XNA residues bearing alternative sugar moieties revealed a strong preference (~5-fold) for 2′-O-methoxyethylribonucleic acid (MOE) (Supplementary Fig. 3). This observation implies that the catalytically active conformation prefers an A-type sugar pucker at position 14, as MOE-modified nucleotides exclusively adopt a 3′-endo sugar conformation due the strong gauche effect imposed by the 2′ substituent. Modest activity gains were also observed for OMe and 2′-fluororibonucleic acid (F-RNA) substitutions, while LNA, FANA, and threose nucleic acid (TNA) substitutions were less active than the native DNA residue, presumably because these residues distort the geometry required for an in-line attack mechanism.

Building from our initial findings, we continued our optimization of Dz 10–23 through modification of the binding arms (Supplementary Fig. 4) as well as a stepwise process of identifying critical residues in the catalytic domain. Although this sequence represents an evolutionary optimum where genetic mutations often lead to variants with reduced catalytic activity[24], very little is known about the functional consequences of chemical substitutions made to the sugar moiety. To investigate such changes, we systematically replaced each residue in the catalytic core with the corresponding OMe nucleotide (for

example, replacing dA with OMe-A). The catalytic profile of the 15 variants uncovered several residues where OMe substitutions led to favorable improvements in catalytic activity (Supplementary Fig. 5). Subsequent evaluation of positions C7, T8, and G14 with various combinations of OMe and MOE chemistry revealed a clear preference for OMe residues at positions 7 and 8, and a MOE residue at position 14 (Supplementary Figs. 6–8). The strong preference for OMe at positions 7 and 8 is consistent with the location of a tight turn in the folded structure of the catalytically active conformation (Supplementary Fig. 1), which lies in close proximity to metal-ion binding site II[22]. Augmenting the triply modified catalytic loop with additional modifications to the binding arms as well as the strategic positioning of 2 phosphorothioate linkages[11] resulted in the generation of 3 highly optimized DNAzyme constructs (variants **38, 42**, and **46**, Supplementary Figs. 4, 9 and 10) that are kinetically indistinguishable in RNA cleavage assays performed with the 10-fold excess substrate (Supplementary Figs. 11 and 12).

To better quantify the activity of our best 10–23 variants, we repeated our kinetic analysis under more stringent conditions in which the reaction mixture contained 100-fold excess substrate. For this experiment, initial rates and catalytic turnover were measured in triplicate for 8 variants spanning the design trajectory from the original all-DNA construct (Dz 1) to the most optimized scaffold (Fig. 1, Supplementary Fig. 11). The resulting data identified variant **46** (Dz 46) as our best design. This construct functions with an initial velocity of ~58 nM/min and is capable of achieving ~65 turnovers in 30 min (Fig. 1). To the best of our knowledge, this level of activity has not been witnessed by any previous RNA-cleaving DNAzyme tested under physiological conditions and compares favorably to the unmodified parent sequence under forcing conditions of 50 mM $MgCl_2$ (pH 8.0)[1]. Importantly, the improvements of Dz 46 over Dz 1 are also transferable to other targets (Supplementary Fig. 13), indicating that it is not a sequence-specific effect. It is also more efficient than a recently described RNA-cleaving XNAzyme evaluated under equivalent conditions (46 turnovers in 96 h)[21] and more efficient than the previously best DNAzyme evaluated against the same RNA target (Supplementary Fig. 14)[25].

Encouraged by the dramatic improvement in multiple turnover activity observed under simulated physiological conditions, we next asked whether the chemically optimized designs functioned with reduced dependency for the divalent magnesium ion cofactor. Single-endpoint detection assays performed across a range of $Mg^{2+}$ concentrations reveal that the best designs exhibit a strongly reduced dependency on the concentration of magnesium present in the reaction mixture than the original all DNA scaffold when tested in the cellular range of 0.1–5.0 mM $MgCl_2$ (Supplementary Fig. 15). In fact, close concordance between the unmodified and modified Dz 10–23 versions does not occur until the reaction mixture reaches a total concentration of 5 mM $MgCl_2$, demonstrating that the design process gave rise to optimized scaffolds with improved magnesium coordination, which increases activity by reducing conformational instability. This observation is supported by competition assays performed in the presence of ATP, which yield only a modest 2-fold loss in activity for Dz 46 (Supplementary Fig. 16) as well as pre-steady state kinetic measurements performed at 0.25 mM $MgCl_2$, yielding a first-order rate constant ($k_{obs}$) of ~0.65 $min^{-1}$ for Dz 46 (Supplementary Fig. 16).

## Substrate specificity of Dz 46

We next asked whether the chemical modifications introduced into the DNAzyme scaffold, which was intended to stabilize the catalytically active conformation of the substrate-bound DNAzyme, might favor the formation of an expanded RNA cleavage motif beyond the commonly discussed purine (R)−pyrimidine (Y) dinucleotide junction[26]. Insights into this question came from the structure of the precatalytic complex of the RNA-bound DNAzyme, which show the enzyme forcing the RNA

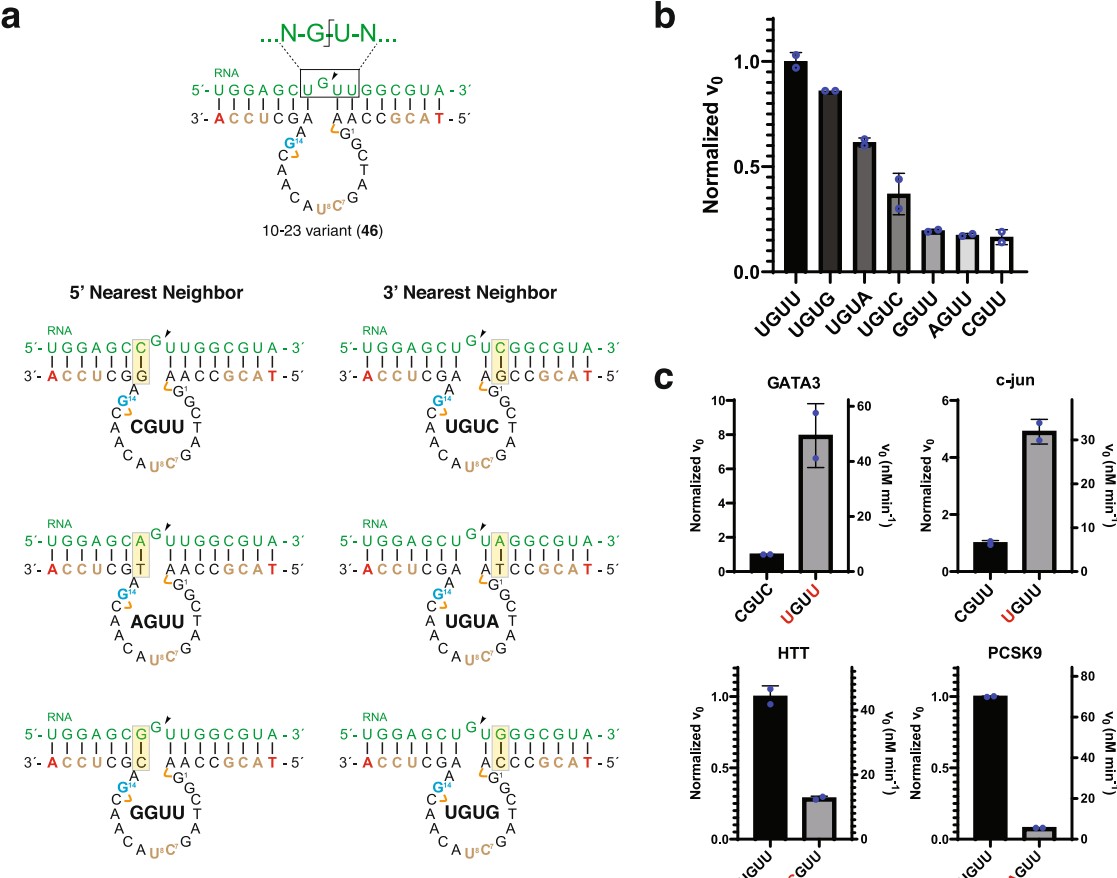

**Fig. 2 | Nearest neighbor analysis uncovers an expanded GU cleavage motif.**
**a** Engineered Dz 10–23_v46 versions and matching substrate designs. Nucleotide positions immediately flanking the 5′ and 3′ sides of the GU dinucleotide cleavage junction were individually varied (yellow boxes) to identify a preferred substrate cleavage motif. The color scheme for DNAzyme-substrate pairs is the same as shown in Fig. 1b. **b** Normalized initial rates observed for each DNAzyme-substrate pair reveal a strong preference for 5′-UGUU-3′ and 5′-UGUR-3′ cleavage sites, where R refers to a purine. **c** Positive (top) and negative (bottom) predictions based on the preferred 5′-UGUU/R-3′ cleavage motifs. Known GATA3 and c-jun cleavage sites

function with inferior activity as compared to their engineered DNAzyme-substrate pairs containing the preferred UGUU motif. Previously untargeted HTT and PCSK9 cleavage sites function with superior activity relative to their low-activity motifs. Mutated nucleotides in the substrate are shown in red. Error bars denote the ±standard deviation of the mean for two independent replicates. Reactions were performed in a buffer containing 1 mM MgCl₂, 50 mM Tris (pH 7.5), 10 mM NaCl, and 140 mM KCl at 37 °C with 1000 nM substrate and 10 nM enzyme (100:1, S:E). Source data are provided as a Source Data file.

substrate to adopt a tight turn at the cleavage junction (Fig. 1)[22]. To explore this question in greater detail, we systematically evaluated the nucleotide position immediately flanking the 5′ and 3′ sides of the preferred G–U cleavage motif using engineered versions of Dz 46 that were programmed to recognize RNA substrates (Fig. 2A) carrying single-nucleotide sequence alterations on the 5′ or 3′ sides of the cleavage junction. Analysis of the initial rates measured for each DNAzyme−substrate pair (Fig. 2B) uncovered a clear preference for a 5′-UGUD-3′ motif (cut site underlined and D corresponds to residues A, G, or U) that was up to 5-fold more active than 5′-XGUU-3′ motifs, where X corresponds to a G, A, or C residue. By comparison, unmodified DNAzymes (Dz 1 variant) exhibit a similar trend when assayed at 5 mM MgCl₂ but show a diminished cleavage site preference for reactions performed at 1 mM MgCl₂ (Supplementary Fig. 17). Importantly, the differences observed between Dz 46-mediated cleavage of the preferred and unpreferred cleavage motifs in short size-matched RNA substrates were maintained in longer RNA substrates (Supplementary Fig. 18), indicating that the trend is not limited to shorter synthetic RNA substrates. However, this trend was not observed for unmodified DNAzymes (Dz 1) against the longer RNA substrates, even at 5 mM MgCl₂ (Supplementary Fig. 19). As such, these results demonstrate the potential for chemically modified DNAzymes to

invade RNA secondary structures that are more likely to arise in the cell.

We validated the positive and negative predictive capabilities of the preferred cleavage motif by evaluating a diverse set of DNAzyme targets. As a test for positive predictive capability, we chose the same GATA-3 (CGUC)[27] and c-jun (CGUU)[28] cut sites previously evaluated in clinical trials as treatments for allergic asthma and nodular basal-cell carcinoma, respectively. Using appropriately designed versions of Dz 46 and short synthetic RNA targets, we compared the cleavage activity of the known cut site to engineered versions carrying the preferred UGUU cleavage motif (Fig. 2C). In both cases, sequence-matched DNAzymes targeting the preferred cut site function with higher activity (5–7-fold) than equivalent DNAzyme-substrate pairs targeting the known cut sites. Likewise, negative predictive capability assays reveal that DNAzymes targeting previously untargeted RNA cut sites identified in Huntington's (HTT, UGUU) and proprotein convertase subtilisin/kexin type 9[29] (PCSK9, UGUU) targets outperform (3–12-fold) engineered cut sites predicted to function with suboptimal activity. The ability to identify optimal G–U cut sites within a disease-associated mRNA target should accelerate future drug discovery efforts with DNAzyme reagents by reducing the number of possible cut sites within a given target and increasing the activity of DNAzymes

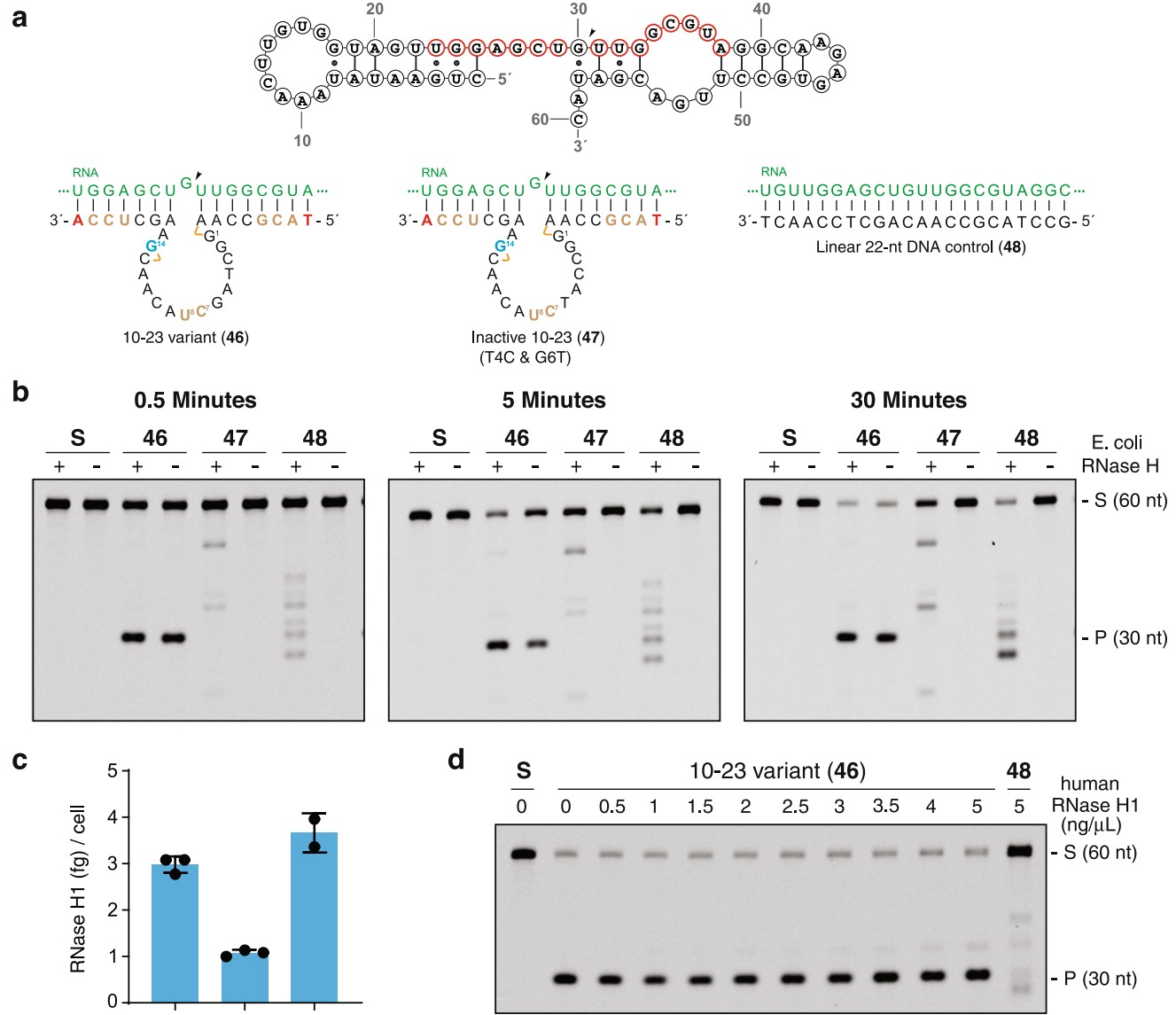

**Fig. 3 | Autonomous DNAzyme-mediated RNA cleavage mechanism. a** Reagents used to evaluate the mechanism of RNA cleavage include a 60 nt RNA segment of KRAS G12V, Dz 10-23_v46 (active Dz), Dz 10−23_v47 (inactive Dz), and a 22 nt linear DNA. The DNAzyme recognition site on the RNA substrate is shown in red, and the RNA cleavage site is indicated with an arrow. The color scheme for DNAzyme-substrate pairs is the same as shown in Fig. 1b. **b** Representative denaturing PAGE gels showing RNA cleavage profiles in the presence (+) or absence (−) of *E. coli* RNase HI (0.05 U/μL) after 0.5, 5, and 30 min of incubation (Dz 46 and 47, *n* = 3; 48, *n* = 1). **c** Quantification of RNase H1 in three mammalian cell lines by ELISA. Cellular levels of RNase H1 range from 0.5 to 2.5 ng/μL. Error bars denote the ±standard deviation of the mean for three independent replicates. **d** Denaturing PAGE gel showing RNA cleavage profiles in the absence (−) or presence (+) of human RNase H1 after a 30 min incubation (*n* = 1). Linear DNA is a positive control. Reactions were performed in a buffer containing 1 mM MgCl$_2$, 50 mM Tris (pH 7.5), 10 mM NaCl, and 140 mM KCl at 37 °C with 250 nM substrate and 250 nM enzyme (1:1, S:E). S: 5′-Cy5-labeled full-length substrate, P: 5′-Cy5-labeled cleavage product. Source data are provided as a Source Data file.

against desired target sites that reside in a favorable UGUD sequence context.

## Autonomous RNA cleavage

Previous studies have shown that RNase H1 contributes to RNA degradation in cultured mammalian cells transfected with unmodified DNAzymes[30]. In such cases, competition exists between the intrinsic activity of the DNAzyme and cellular RNase H1, which recognizes the substrate binding arms of the DNAzyme as antisense oligonucleotides. Although chemical modifications have reduced RNase recognition[17], this topic continues to be discussed by DNAzyme development laboratories[31]. We postulated that the chemical modifications introduced into the binding arms of the 10−23 scaffold, coupled with the

substantial multiple-turnover activity observed under near-physiological conditions, would render Dz 46 less susceptible to the effects of an RNase H-induced cleavage mechanism. To investigate this possibility, we evaluated the catalytic activity of Dz 46 against a synthetic 60 nt RNA substrate in time-course experiments that either contained or lacked recombinant *E. coli* RNase HI (Fig. 3A). Analysis of the resulting denaturing PAGE gels indicate that the reaction produces the desired site-specific cleavage product regardless of whether RNase H is present or absent from the reaction mixture (Fig. 3B). By contrast, control experiments performed using a catalytically inactive version of Dz 46 and a linear antisense DNA oligonucleotide (ASO) yield cleavage patterns that are dependent upon the presence of RNase HI. The striking difference in behavior between the active DNAzyme and the

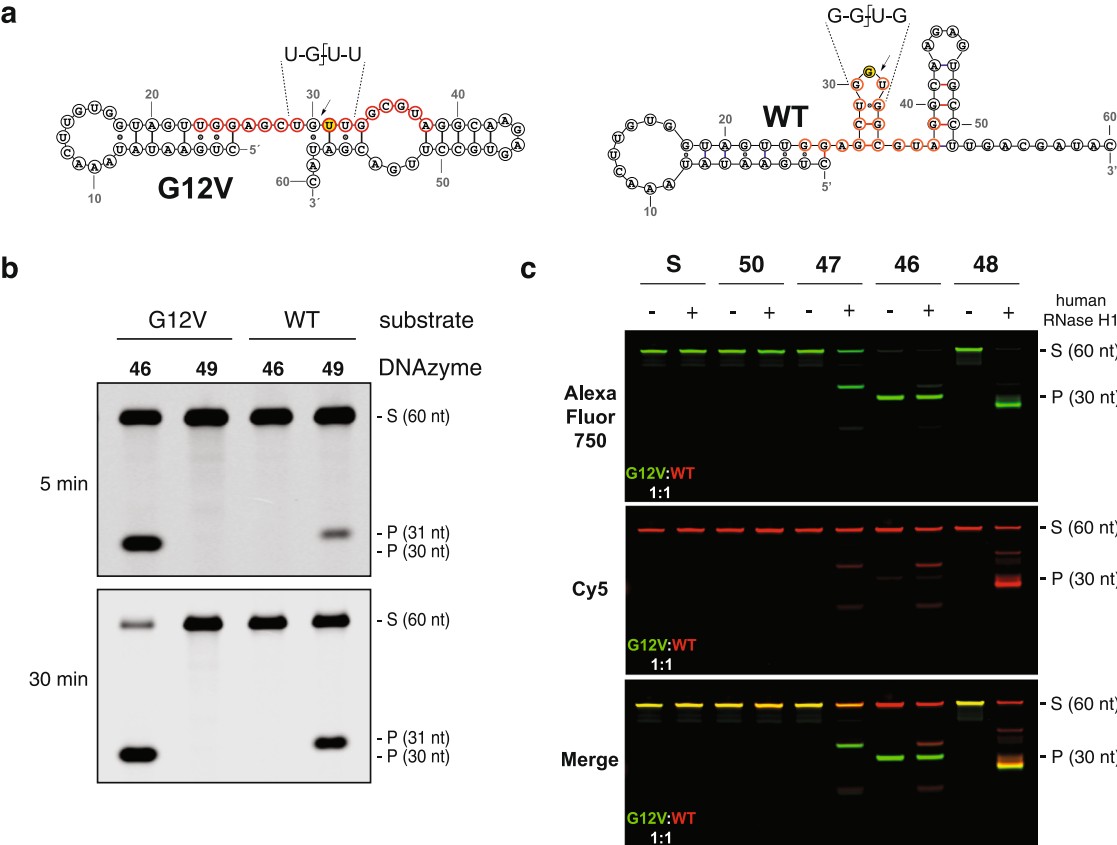

**Fig. 4 | Allele-specific targeting of wild-type and KRAS G12V RNA under cell-free conditions. a** Substrates used to evaluate allele-specific RNA cleavage of KRAS G12V include two 60 nt segments of KRAS mRNA that differ by a single point mutation. The G12V version carries a G-to-U mutation (yellow) that leads to a glycine-to-valine mutation in the translated KRAS protein. RNA binding sites targeted by the wild-type (49) and G12V (46)-specific DNAzymes of Dz 10-23_v46 are shown in red, with the cleavage site indicated with an arrow. **b** Denaturing PAGE gel showing allele-specific RNA cleavage of both alleles by the wild-type (49) and G12V (46)-specific DNAzymes after 5 and 30 min (*n* = 3). Reactions were performed in a buffer containing 1 mM MgCl₂, 50 mM Tris (pH 7.5), 10 mM NaCl, and 140 mM KCl at

37 °C with 500 nM substrate and 50 nM enzyme (10:1, S:E). S: 5′-Cy5-labeled full-length substrate, P: 5′-Cy5-labeled cleavage product. **c** Denaturing PAGE gel showing RNA cleavage profiles in reactions containing both the wild-type (red) and G12V (green) substrates (1:1) in the absence (−) or presence (+) of 5 ng/μL human RNase H1 after a 30-min incubation (*n* = 1). Individual Cy5 (red) and AlexaFluor750 (green) channels, as well as the merged image, are shown. Reactions were performed using unpaired Dz (50), inactive Dz (47), active G12V Dz (46), and 22 nt linear DNA control (48) in buffer containing 1 mM MgCl₂, 50 mM Tris (pH 7.5), 10 mM NaCl, and 140 mM KCl at 37 °C with 250 nM total substrate and 250 nM enzyme. Source data are provided as a Source Data file.

control reagents indicates that Dz 46 is sufficient and necessary to produce the predicted RNA cleavage product, supporting the hypothesis that Dz 46 functions by an autonomous site-specific DNAzyme-mediated RNA cleavage mechanism.

Since our initial assays were performed using the manufacturer-recommended concentration of 0.05 units/μL of *E. coli* RNase HI, which is presumably required to facilitate efficient molecular biology transformations, we chose to examine the mechanism of DNAzyme cleavage under conditions that were more biologically relevant. We began by quantifying the cellular concentration of RNase H1 in cultured HEK293, K562, and NCI-H441 cell lines. Using an ELISA assay, we found that the cellular concentration of RNase H1 ranged from 1.0 to 3.5 fg/cell (Fig. 3C), which translates to ~0.5–2.5 ng/μL (Supplementary Table 5). Relative to the linear DNA control, DNAzyme-mediated RNA cleavage assays performed in the physiological range of human RNase H1 produce the desired site-specific cleavage product (Fig. 3D). This observation strengthens the hypothesis that Dz 46 functions as an autonomous gene silencing reagent independent of RNase H1 engagement.

**Allele-specific RNA cleavage**

In contrast to protein-based gene silencing technologies, the mechanism of DNAzyme-mediate strand cleavage allows for precise targeting of specific nucleotide mutations (SNPs) responsible

for disease-associated phenotypes[2]. A classic example is KRAS—a master regulator of cell growth and division with activator mutations found in ~25% of all human cancers[32]. One commonly studied KRAS mutation is G12V, caused by a G→U mutation in position 2 of codon 12, which leads to glycine (G) to valine (V) mutation in the amino acid sequence[33]. Unlike G12C, the G12V mutation is viewed as an undruggable target as the valine residue lacks a nucleophilic moiety for targeting by small molecule inhibitors[34,35]. We assessed the ability of Dz 46 to distinguish allelic mutations in synthetic KRAS RNA substrates represented by a 60 nt segment of the mRNA (Fig. 4A). DNAzymes engineered to recognize the wild-type and mutant KRAS substrates functioned with >99% specificity (Fig. 4B), as no cross-reactivity was observed for either DNAzyme against the opposing substrate. Consistent with our substrate specificity study, Dz 46 is ~5-fold more active than Dz 49, which is due to the presence of a favorable UGUU cleavage motif in the mutant substrate and a less favorable GGUG motif in the wild-type substrate.

Next, we examined the potential for allele-specific strand cleavage under competitive conditions in which both allelic substrates are present in the reaction mixture. This assay is more representative of the cellular environment and includes reactions that were performed in the presence and absence of human RNase H1 poised at a concentration that is 2-fold above the cellular range. To distinguish the

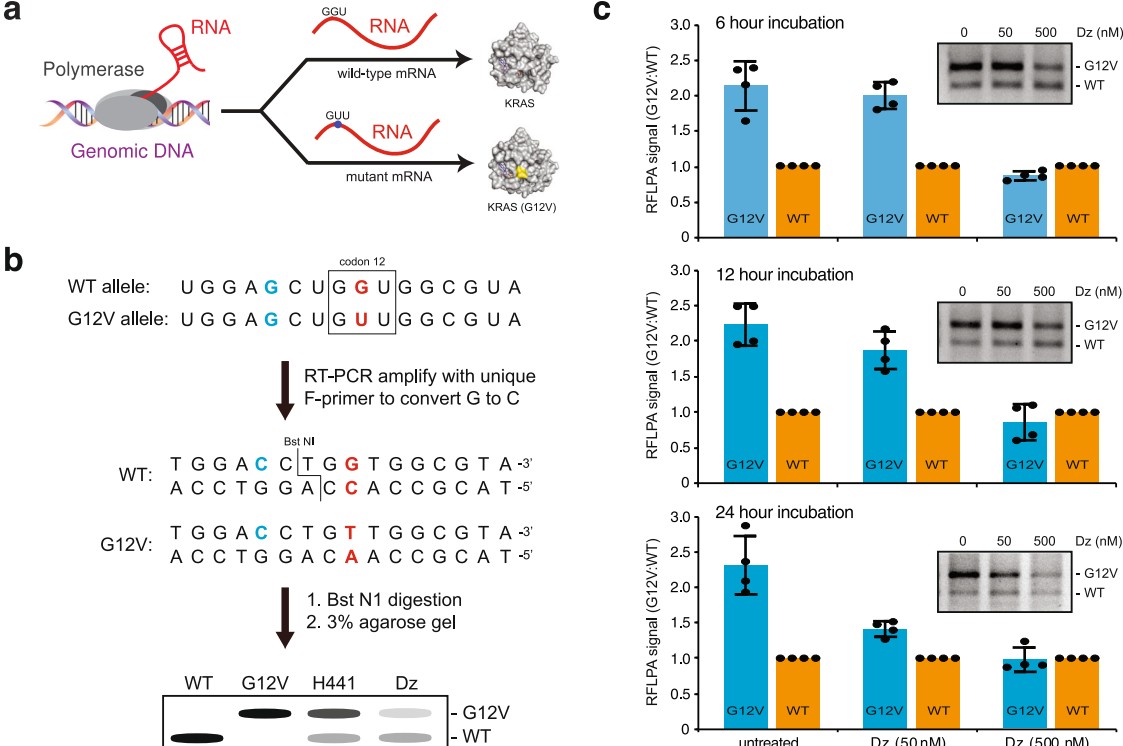

**Fig. 5 | Allele-specific targeting of endogenous KRAS G12V RNA in a lung cancer cell model. a** Schematic representation of the transcription and translation of allelic mutations associated with the expression of wild-type and G12V mutant KRAS proteins in NCI-H441 cells. **b** Schematic view of the PCR restriction fragment length polymorphism assay (PCR-RFLP) used to distinguish KRAS alleles associated with wild-type and mutant G12V protein expression. G-to-C mutation introduced by primer during PCR is shown in blue. The second position of codon 12 (boxed) encoding G12V or WT KRAS is shown in red. **c** Quantification of allele-specific G12V knockdown levels observed in NCI-H441 cells after 6, 12, and 24 h of treatment with either 50 or 500 nM Dz 10−23_v46 targeting KRAS G12V mRNA. Representative PCR-RFLP results are provided as an insert for each plot (*n* = 4). Error bars denote ± standard deviation of the mean for *n* = 4: 2 biological and 2 technical replicates. Source data are provided as a Source Data file.

wild-type and mutant substrates, the synthetic RNA strands were prepared with 5′-modified Cy5 (red) and Alexa Fluor 750 (green) tags, respectively. In these assays, Dz 46 exclusively cleaves the mutant KRAS substrate; however, a trace amount of RNase H1-mediated cleavage is observed for the wild-type substrate, which is attributed to the formation of a stable but inactive complex between the mutant Dz and wild-type substrate (Fig. 4C and Supplementary Fig. 20). Standard positive controls performed with inactive DNAzyme and the ASO yield the expected RNase H1 cleavage products, while the non-binding DNAzyme and substrate only controls show no signs of RNA cleavage (Fig. 4C).

Having demonstrated that Dz 46 is capable of allele-specific RNA cleavage, we next asked whether similar effects could be achieved for a high-value endogenous mRNA target. For this study, we chose NCI-H441 as a model adenocarcinoma cell line expressing 3−4 genomic copies of G12V to wild-type KRAS per cell (Fig. 5A). We used a PCR restriction fragment length polymorphism assay (PCR-RFLP) to distinguish cleavage between the two allelic mRNA transcripts (Fig. 5B). With this technique, RT-PCR is used to introduce a mutation into the amplicons that lead to the formation of a restriction site in the wild-type sequence only. Following restriction enzyme digestion, allele-specific gene knockdown activity is assessed by quantifying the ratio of the mutant and wild-type bands by agarose gel electrophoresis. Our results indicate that modest mRNA knockdown of the mutant gene is observed after 24 h of treatment with 50 nM DNAzyme (Fig. 5C). Higher dosing with 500 nM DNAzyme leads to rapid and sustained knockdown of mutant KRAS G12V mRNA, as observed between 6 and 24 h of treatment (Fig. 5C, Supplementary Fig. 21). This result compares favorably against that of X10−23 targeting the same G12V

mutation, which required 96 h of treatment to observe similar levels of allele-specific gene silencing[37].

## Discussion

The site-specific cleavage of a target RNA molecule by a natural or in vitro evolved nucleic acid enzyme is one of the best-characterized reactions in nucleic acid chemistry[38,39]. In fact, the mechanism of action and strategy for evolving these reagents is so well understood that this functional paradigm has been extended to XNAzymes, whose backbone structures encompass synthetic forms of nucleic acid polymers composed entirely of non-natural sugar-phosphate linkages[19−21]. However, despite major growth in this area of synthetic biology, the ability to generate nucleic acid enzymes with robust multiple turnover activity under physiological conditions has proven more challenging than expected. Even Darwinian evolution experiments aimed at querying large combinatorial libraries for individual sequences that function with reduced magnesium dependency have fallen short of expectations[13]. Given the significant effort expended toward this problem, one could conclude that the chemical repertoire available to natural genetic polymers is insufficient to achieve the level of activity required for clinical applications.

In the current study, we evaluated the potential for chemical evolution to overcome this barrier by augmenting a known DNAzyme with additional functional properties. We postulated that the substitution of natural DNA residues for synthetic congeners at key structural positions in a DNAzyme scaffold might favor the formation of a catalytically active conformation by shifting the equilibrium of folded structures away from the ensemble of alternative yet unproductive conformations. Through iterative cycles of design, we

uncovered permissive sites in the 10–23 DNAzyme scaffold where chemical modifications afforded clear stepwise improvements in multiple turnover activity under near-physiological conditions. This effort culminated in the discovery of Dz 46, a chemically engineered 10–23 variant that functions with unprecedented multiple turnover activity under experimental conditions simulating the cellular environment. Biochemical characterization indicates that the heavily modified DNAzyme achieves heightened catalytic activity through improved cofactor binding or coordination, which is required for proper folding and function of the DNAzyme. We suggest that the approach taken was sufficiently general that it could be applied to other DNAzymes, especially those whose structures have been determined. While insights from the NMR-based structure of 10–23 helped to guide the design of Dz46, other structural insights from RNA-cleaving nucleic acid enzymes that share a common active site motif known as the L-platform, may facilitate the design of other DNAzymes, such as 8–17[40]. It should also be noted that RNA and DNA-catalyzed RNA-cleavage mechanisms described by Breaker and others may shed light on other potential design principles for engineered DNAzymes[41].

One interesting result to come from our study was the finding that Dz 46 favors a highly specific tetranucleotide cleavage motif (UGUD). This observation is likely due to the increased structural rigidity of the DNAzyme scaffold caused by the introduction of diverse chemical modifications throughout the catalytic loop and binding arms of the molecule. Elucidation of the preferred cleavage motif, which was validated using both positive and negative predictive agreement assays across four different disease-associated targets, offers a streamlined approach for designing new 10–23 variants for clinical applications as it reduces the number of G–U cut sites in an mRNA target to a subset that reside in a preferred sequence context. We note, however, that the additional constraints imposed by the preferred cleavage motif could restrict the number of targets amenable to Dz cleavage. Another striking feature of our study was the discovery that Dz 46 functions as an autonomous reagent under physiologically relevant concentrations of RNase H1, which were empirically determined using an ELISA assay. We postulate that the fast catalytic rate, coupled with the presence of chemical modifications in the substrate binding arms, greatly reduces the potential for RNase H1 engagement.

One area of biomedicine where DNAzymes, like Dz 46, could have a major impact on human health is in the validation and treatment of allele-specific genetic diseases[42]. Currently, protein-based gene silencing reagents struggle to distinguish SNPs in an RNA target[43,44], while DNAzymes can unequivocally target mutant and wild-type alleles with high efficiency and precision. In this way, Dz 46 offers an additional tool in the medicinal chemistry toolbox that allows for the precision cutting of an allelic mRNA target in cultured mammalian cells. Recognizing that >200,000 known acquired GU mutations can be found in the Cancer Genome Atlas database[37], it is reasonable to assume that Dz 46 is well poised to expand the concept of allele-specific gene silencing.

In summary, we applied a chemical evolution approach to discover a highly modified 10–23 DNAzyme variant that functions with unusually high multiple turnover activity under near-physiological conditions. The generality of our approach opens the door to other examples of DNAzymes that could be improved for future biomedical applications.

## Methods
### Materials
Oligonucleotide sequences used in this study are listed in Supplementary Tables 1–4. All oligonucleotides were purchased from Integrated DNA Technologies (Coralville, Iowa), except for Dz 6 and Dz 7, which were prepared by solid-phase oligonucleotide synthesis using an ABI 3400 automated DNA synthesizer. The TNA-G phosphoramidite used for Dz 7 was chemically synthesized as described previouly[45]. All other reagents used for solid-phase synthesis were purchased from Glen Research. In-house oligonucleotides were synthesized in DMT-on mode, desalted, and purified by reverse-phase HPLC. Oligonucleotides purchased from IDT were used directly without further purification. Stock solutions used to prepare reaction buffers were purchased as follows: 1 M Tris-HCl pH 7.5 (Quality Biological), 1 M MgCl₂ (Invitrogen), 2 M KCl (Invitrogen), and 5 M NaCl (Sigma). Molecular and cellular biology reagents include Dulbecco's modified eagle medium, DMEM (ThermoFisher Scientific), RPMI 1640 medium, and NCI-H441 lung adenocarcinoma (ATCC, Cat#: ATCC-CRM-HTB-174™), jetPRIME® DNA and siRNA transfection reagent (Polyplus Transfection), Trizol reagent (Invitrogen). RNA clean & concentration-5 kit (Zymo Research), Turbo DNase and SuperScript III First-strand Synthesis System (Invitrogen-Life Technologies, CA), *E. coli* RNAse HI and BstNI (NEB), Human RNAse H1 (Abcam, Cat# Ab153634). Bichinconinic Acid (BCA) assay (GBioSciences), Human Ribonuclease H ELISA kit (AssayGenie, Cat# HUDL02549)

### In vitro cleavage reaction
Totally, 10–23 variants were assayed under multiple turnover conditions using either 10:1 (S:E, 500 nM RNA substrate and 50 nM DNAzyme) or 100:1 (S:E, 1000 nM RNA substrate and 10 nM DNAzyme) in simulated physiological reaction buffer containing 50 mM Tris-HCl (pH 7.5), 10 mM NaCl, 140 mM KCl, and 1 mM MgCl₂, unless otherwise noted. DNAzymes and substrates were annealed in the absence of MgCl₂ by heating for 5 min at 95 °C and cooling for 5 min at 4 °C. Solutions were equilibrated to 37 °C for 2 min before initiating the reaction with the addition of MgCl₂ and kept at 37 °C throughout the time course. Aliquots (1.5 μL) of the reaction mixture were removed at the indicated time points and quenched in 16.5 μL stop buffer (99% deionized formamide, 25 mM EDTA). Quenched samples were denatured at 95 °C for 10 min and analyzed by 20% denaturing PAGE. Gels were imaged using an Odyssey CLx imaging system (LI-COR) and quantified using Image Studio Lite (LI-COR).

### Initial velocity ($v_0$) determination
Initial velocity ($v_0$) was measured between 0 and 30 min under 100:1 (S:E) multiple turnover conditions as described above. Values for $v_0$ were calculated from a linear fit of the first 10–15% of the cleavage reaction using Eq. (1)

$$v_0 = \frac{\left(Y_f - Y_0\right) \times [S_0]}{(t_f - t_0)} \tag{1}$$

where $v_0$ is the initial velocity (nM*min⁻¹), $Y_f$ is the percent cleavage at finite time $t$ when the first 10–15% cleavage is reached, $Y_0$ is the percent cleavage at $t_0$, and $[S]_0$ is the initial substrate concentration (nM) at $t_0$.

### Rate constant ($k_{obs}$) determination
Single turnover experiments were performed at 37 °C under pseudo-first order reaction conditions using a 1:5 ratio of S:E (0.5 μM substrate and 2.5 μM enzyme) in a buffer containing 50 mM Tris-HCl (pH 7.5), 10 mM NaCl, 140 mM KCl, and 0.25 mM MgCl₂. Aliquots of the reaction mixture were collected and resolved by PAGE, as described previously. Observed rate constants were obtained by fitting the percentage of the cleaved substrate over the reaction time (min) to the one-phase association Eq. (2) using Prism 9 (GraphPad, USA)

$$Y = Y_0 + (Y - Y_0) \times \left(1 - e^{-k_{obs} \times t}\right) \tag{2}$$

where $Y$ is the percent cleavage at finite time $t$, $Y_0$ is the percent cleavage at $t_0$, $Y_\infty$ is the percent cleavage at an infinite time where the reaction plateaus and $k_{obs}$ is the observed pseudo-first-order rate constant (min⁻¹).

### Data analysis software

Prism 9 (Graphpad, USA) for curve fitting and Microsoft Excel was used to generate plots.

### Mg²⁺ dependency

$Mg^{2+}$ dependency was measured under multiple turnover conditions as described above using 100:1 (S:E) in cleavage reactions containing defined concentrations of $MgCl_2$ (0, 0.10, 0.25, 0.50, 1.0, 2.5, and 5.0 mM). After 20 min of reaction, aliquots (1.5 μL) were removed, quenched, and analyzed by 20% denaturing PAGE as described in the in vitro cleavage reaction section. Similar assays were performed in the presence of 1 mM ATP to assess the cleavage activity of Dz 46 in the presence of a natural $Mg^{2+}$ chelator.

### Substrate specificity analysis

To determine the preferred substrate cleavage motif, a nearest neighbor analysis was performed by measuring the initial rates for matched Dz-substrate pairs containing single nucleotide changes to the 5′ and 3′ positions adjacent to the GU cleavage site in the RNA substrate. Reactions were performed as described in the initial rate section above. For each Dz-substrate pair, the initial velocity was calculated and normalized to the fastest motif.

### RNase H1 evaluation

The contribution of RNase H toward substrate cleavage was evaluated using *E. coli* RNase HI and human RNase H1. For the *E. coli* RNase HI assay, the long (60 nt) G12V KRAS RNA substrate (250 nM) was incubated at 37 °C for 10 min in the presence of 50 mM Tris-HCl (pH 7.5), 10 mM NaCl, 140 mM KCl. Reactions were initiated with a final concentration of 1 mM $MgCl_2$, 0.05 U/μL *E. coli* RNase HI or water for RNase H free controls, and 250 nM of either Dz 46 (active), Dz 47 (inactive), or a linear 22 nt all-DNA ASO. Aliquots (1.5 μL) were removed at 30 min, quenched, and analyzed by 8% denaturing PAGE as described above. The human RNase H1 assay was performed similarly to the *E. coli* assay, with the exception that human RNase H1 levels spanned the biological range of 0–5 ng/μL, as determined by ELISA (see below).

### Allele-specific RNA cleavage in vitro

Cell-free allele-specific cleavage assays were performed using the long (60 nt) versions of the wild-type and G12V KRAS RNA substrate (250 nM) in two formats: alone or in competition with one another.

Individual format. DNAzyme targeting specificity was assessed using multiple turnover conditions (10S:1E): 500 nM of the long (60 nt) versions of either WT or G12V KRAS RNA substrate was mixed with 50 nM Dz 49 (WT, active) or Dz 46 (G12V, active) in the buffer containing 50 mM Tris-HCl (pH 7.5), 10 mM NaCl, 140 mM KCl, 1 mM $MgCl_2$ and incubated at 37 °C. Aliquots of 1.5 μL reaction mixture were taken at 5 min and 30 min, quenched, and resolved using 8% denaturing PAGE as described in the in vitro cleavage reaction section.

Competition format. Cleavage activity was assessed using 250 nM of Dz 46 (active), Dz 47 (inactive), Dz 50 (active, non-binding), or ASO (48) and mixed with two 60 nt RNA substrates in a 1:1 ratio: 125 nM WT KRAS with a 5′Cy5 tag and 125 nM G12V KRAS with a 5′Alexa Fluor 750 tag. Reactions were performed at 37 °C in buffer containing 50 mM Tris-HCl (pH 7.5), 10 mM NaCl, 140 mM KCl, and 1 mM $MgCl_2$, with or without 5 ng/μL human RNase H1. Aliquots (1.5 μL) were removed after 30 min, quenched, and resolved using 20% denaturing PAGE as described in the previous section.

### Intracellular assays

*Cell lines and mammalian cell culture conditions.* NCI-H441 (Cat. # CRM-HTB-174), HEK293 (Cat. # CRL-1573), and K562 (Cat. # CCL-243) cell lines used in this study were purchased from American Type Culture Collection (ATCC). HEK293 cells were cultured in DMEM. K562 and NCI-H441 cells were cultured in RPMI 1640. Media were supplemented with 10% fetal bovine serum and 1% penicillin-streptomycin (1 mg/mL) and grown at 37 °C, 5% $CO_2$.

*Enzyme-linked immunosorbent assay (ELISA).* HEK293 and K562 cells (3 aliquots/cell line) and NCI-H441 cells (2 aliquots) were harvested, counted, and re-suspended in 1× PBS pH 7.4 supplemented with DNase, RNase A, and proteinase inhibitor cocktail set VII and subjected to protein extraction and BCA for quantification. *Hs*RNase H1 concentration from the total protein of the three cell lines was determined using the human ribonuclease H ELISA kit according to the manufacturer's instruction and quantified using a Clariostar Plus microplate reader (BMG Labtech, NC).

*Transfection.* After 48–96 h post-seeding and when cell density reached 60%, NCI-H441 cells in 6-well plates were transfected with transfection carrier only (0, Negative control), 50 or 500 nM of Dz 46 using JetPrime transfection reagent according to manufacturer's instruction except for a higher amount (5×) of transfection reagent. For negative controls, the volume of JetPrime reagent used for each well was the same as those with DNAzymes to ensure the same transfection condition in the control and experimental samples. At 6, 12, and 24 h post transfection, cells were harvested and subjected to total RNA extraction.

*RNA isolation and reverse transcription (RT).* To each well of a 6-well plate, total RNA isolation was performed using 1 mL/well Trizol reagent according to the manufacturer's instructions. Total RNA was treated with Turbo DNase (20 U/reaction) at 37 °C for 30 min with shaking, followed by purification using RNA clean and concentration-5 kit according to manufacturer's instructions. Total RNA was eluted twice with 22 μL per elution, resulting in 44 μL total volumes. DNA-free RNA (700 ng) was subjected to cDNA synthesis using SuperScript III First-strand Synthesis System with random hexamer primers in a 20 μL reaction according to the manufacturer's instructions. Template mRNA in the RNA:cDNA hybrid was removed by RNase H (2U/cDNA reaction) digestion resulting in a final volume of 21 μL.

*Restriction fragment length polymorphism assay (PCR-RFLPA).* PCR-RFLPA was performed as previously described with the following changes[37]. In brief, upon cDNA synthesis, amplification of 200 nt KRAS fragment was performed using a unique sense primer to introduce a single substitution (G to C) at the first nucleotide (bold) of KRAS codon 11 (**G**CT to **C**CT). This substitution introduces *the Bst*NI recognition site ("CCWGG") spanning from codon 11 (**C**CT) to codon 12 (GGT) of the KRAS WT allele only. G12V KRAS allele has a mutation (bold) at codon 12 (G**G**T to G**T**T) and thus resists *Bst*NI cleavage. The antisense primer is oKN483R: 5′-TCCTCTTGACCTGCTGTGTCG-3′, and the sense primer is oKN484F: 5′- TTATTATAAATAATGACTGAATATAAACTTGTGGTAG TTGGA**c**CT-3′ with a lower case "c" denotes a substitution. 5 μL of unpurified cDNA was used as a template for PCR (40 cycles), and the amplicon was subjected to purification. 500 ng of the amplicon was digested with 20 U *Bst*NI for 1 h at 60 °C in a 50 μL reaction. The G12V allele remained intact (200 nt). WT allele was cleaved and resulted in two fragments: 150 nt and 50 nt. Upon resolving in a 3% agarose gel (132 ng/lane), WT and G12V alleles were separated by a difference of 50 nt. For loading control, 1 μL of unpurified cDNA was used as a template for GAPDH amplification (PCR: 20 cycles). Amplicons were purified using Zymo DNA-Clean and Concentration-5 kit (Zymo, Cat# D4003) and eluted with nuclease-free water twice (22 μL/each). GAPDH primers are oKN60F: 5′-ACCATCTTCCAGGAGCGAGATCCCTC-3′ and oKN61R: 5′-TGCAGGAGGCATTGCTGATGATCTTGA-3′.

### Statistics and reproducibility

Descriptive statistics were calculated in Microsoft Excel. Experiments were generally reproducible. No statistical method was used to pre-determine the sample size. During revision, the data shown in Fig. 2c and Supplementary Fig. 16a were replaced in their entirety for consistency. The Investigators were not blinded to allocation during experiments and outcome assessment.

**Reporting summary**

Further information on research design is available in the Nature Portfolio Reporting Summary linked to this article.

## Data availability

The authors declare that the data supporting the findings of this study are available within the article and its Supplementary information file. Structural information referenced here can be found in the Protein Data Bank under accession code 7PDU. The raw data generated in this study are provided in the Supplementary Information/Source Data file. Source data are provided in this paper.

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

## Acknowledgements

This work was supported by 1E Therapeutics.

## Author contributions

J.C., K.N., and T.M. conceived the project and designed the experiments. K.N. and T.M. performed the experiments. J.C. wrote the paper with drafts from K.N. and T.M. All authors reviewed and commented on the paper.

## Competing interests

The authors (J.C., K.N., and T.M.) and the University of California-Irvine have filed provisional patent applications on the chemical composition and cleavage preference of Dz 46. J.C. is a consultant for 1E Therapeutics.
