## [Peer Review File · Nature Communications]

Reviewers' Comments:

Reviewer #1:

Remarks to the Author:

The manuscript by Nguyen et al. presents the systematic development of new DNAzyme (Dz) variants using a comprehensive set of chemical modifications. The approach converges on a greatly improved new variant (Dz 46) that shows significantly more activity under near physiological conditions. In addition, specific features of Dz 46 are presented, including an extended cleavage site preference, activity in the presence of RNase H1, and allele specificity in cell-free and cell-based KRAS model systems. Overall, the manuscript is of very high scientific quality and presents very important new results that will have a strong impact in the field of RNA-cleaving XNAs and could further advance the development of Dz-based therapeutics. I have only a few comments that should be clarified to further improve the manuscript.

Moderate Points:

1. Fig. S15A: In general, the ATP competition test is very interesting, however, the margin of error in the data presented seems to be quite large, which (partially) prevents a clear conclusion. If possible, the data and its interpretation would benefit from more data points (replicates) to reduce the margin of error. This could help to strengthen this important aspect.
2. Fig. 2: Since the effects of the cleavage site neighbours differ from previous results, it would be interesting to include comparative data on the Dz 1 variant to evaluate whether this is caused by the chemical modifications or is "just" selective for the sequences used.
3. Fig. S16 B: Since a longer RNA target has lower activity (as expected), it would be interesting as well to see how Dz 1 (and possibly Dz 4) compares.
4. Fig. 2C: While it is interesting that the cleavage-site site preference is transferable to other systems, it would be equally interesting to see if the increased activity is also transferred to these systems. Therefore, please include the absolute values of the velocities already recorded (e.g., using a second y-axis). In addition, it would be interesting to see if the improvements of Dz 46 are also generally transferable to these systems, e.g., by comparison with the respective Dz 1 system (e.g., fold change in relative activity).
5. Fig. 3: The data shown in Fig. 3 are "good to have," but they do not add too much value to the manuscript. In this regard, the data in Fig. 4C are much more relevant. Also, the discussion of these data can be considerably shortened, e.g., autonomous cleavage has also been shown in several previous assays using a longer RNA substrate. I would therefore recommend moving Fig. 3 to the SI. Instead, I would recommend moving some data of the most central modifications from the SI to the main text.
6. In the RNase experiments it would, however, be good to also have data with a ratio of 10:1 (RNA:Dz), as the different turnover conditions might affect RNase differently (as already potentially seen in Fig. 4C).
7. Fig. 5C: It would be helpful to also plot/discuss absolute values to e.g., see changes in wt levels (it appears that this is affected at 24 h and would be relevant for therapeutic applications).
8. It should be briefly discussed that the stabilization of the binding arms due to the chemical modifications alters the dissociation temperatures of the respective Dz:RNA complexes (pre- and post-catalytic), thus favoring in particular the multiple turnover conditions of the new variants over the Dz 1 variant and all other variants without modifications in the arm regions, especially Dz 4 (when using assay temperatures that are not ideal for Dz 1/4 turnover). The authors should either briefly discuss this aspect and/or test the Dz 1/4 variant at different temperatures to find their optimal turnover condition.

Minor points:

9. Fig. S4: In the main text Fig. S4 should be explained/mentioned before the results of Fig. S5 are discussed.

10. Fig. S5: Unlike most other plots, it appears that these data do not allow a firm conclusion as to whether the observed slightly increased/decreased activities are truly significant. However, this set of experiments does a good job of identifying modifications that render the Dz inactive and modifications that keep it active (with roughly comparable activities). Since the subsequent figures break down the respective modifications, I would recommend interpreting the data of Fig. S5 in the text as yes/no only.

11. The Usage of 'OMe' is not consistent in the supplement (e.g., Fig. S8A Table 'OME', and several other positions). Please crosscheck and use consistent nomenclature.

12. Is there a reason why variant 44 was not evaluated further? It does not seem to differ from the three selected variants.

13. It would be helpful to include a table of the key modifications/constructs in Fig. 1 (maybe remove gels in Fig. 1C to SI).

14. The way the cleavage velocity is calculated (i.e. using 10-15% cleavage product), could potentially change between the assays using 10:1 and 100:1 (RNA:Dz) ratios. How does the velocity compare between the data shown in Fig. S12B,C vs. Fig. 1D,E ?
If this is an issue it should be briefly discussed and/or a more robust analysis used.

15. Please rewrite phrase '10-30 fold less dependent' (page 7) to e.g., 'strongly reduced dependency on Mg²⁺' (since the data do not quantify the 'dependency').

16. The interpretation of improved cofactor (Mg²⁺) binding due to the modifications is speculative. The modifications could also render cofactor binding obsolete at some of the binding sites. Please rewrite the respective sentences accordingly.

17. The authors highlight the additional constrains of the cleavage site as an exclusively positive effect. While reduced number of target sites will indeed simplify the selection process, it will also impose additional constrains. Please rewrite the respective sentences accordingly.

18. While the new variants are certainly impressively effective, phrases like 'making it the fastest' are still difficult to assess, since most previous Dz variants have been investigated under different conditions. In addition, regular 10-23 Dz (Dz 1) variants with considerably higher activity as compared to the used Dz 1 (KRAS) variants have been reported, making the comparison even more difficult. Therefore, please use a less 'absolute' wording.

19. I would recommend not selecting single parameters, such as ">250 times more effective than" when making quantitative comparisons with results from other groups; this oversimplifies "efficiency." In this example, the discussed variants indeed show significantly reduced multiple turnover rates when using minimal RNA substrates; however, they do not seem to suffer as much when using longer RNA constructs and also show comparable effects to Dz 46 in cellular assays. Therefore, please either include a more detailed discussion or write the relevant statement more generally.

20. While the used buffer is well selected and reflects ion concentrations normally found under physiological conditions, many other factors will affect the in vivo activity of Dz. The authors already demonstrate that 1 mM ATP (which is still below frequently found cellular concentrations) will reduce activity by 50% and many other molecules may also interfere with Mg²⁺ and/or the DNA/RNAs. Therefore, please rephrase all sentences referring to in vitro conditions from "under physiological conditions" to "under near physiological conditions"

Reviewer #2:

Remarks to the Author:

The authors describe the strategic chemical evolution of the RNA-cleaving 10-23 DNAzyme for improved catalytic activity under physiological conditions. As the authors describe, achieving high-activity DNAzyme function *in vivo* has been a long-standing challenge, due to the generally high divalent metal ion (Mg²⁺) requirement of DNAzymes as well as the alternative RNase H-based antisense mechanism enabled by the Watson-Crick DNAzyme binding arms. Here they use the high-resolution structural information published by others in ref. 21 (Nature 2022) to design a stepwise series of chemical changes in 10-23, eventually finding DNAzyme 46 (Dz 46). They show that Dz 46 has the highest reported multiple turnover of any DNAzyme for RNA cleavage under physiological conditions. The data appear strong to support an autonomous, RNase H-independent cleavage mechanism. Allele-specific RNA cleavage is established, including in a cellular assay by PCR-RFLP analysis. I support publication in Nature Communications after suitable revisions to address the following comments.

1. The claim of "perfect specificity" (page 10) of the DNAzyme for the two KRAS mRNA targets can surely be quantified more precisely, based on the experimental upper limit of "no cross-reactivity was observed for either DNAzyme against the opposing substrate". For instance, if the detection limit in Figure 4B is <1%, then the specificity would be >99%. Any such value should be corrected for the % yield in the relevant active lane (e.g., <1% versus 50% yields would mean >98% specificity).

2. "We suggest that the approach taken was sufficiently general that it could be applied to other DNAzymes that have been developed..." (page 12). This sounds right, but only as long as those other DNAzymes also have the detailed structural information analogous to that for 10-23 in ref. 21 - which I understand took those authors about a decade to obtain! I think this warrants a disclaimer in the main text that application to other DNAzymes will require structural information that is not trivial to obtain.

3. It is surprising not to see any direct citation of the extensive work of David Perrin (U. British Columbia) on seeking metal-independent RNA-cleaving DNAzymes. There is a sort of hint of this in the citation of ref. 12, which is a review from the authors themselves rather than Perrin. The authors can correctly note that Dz 46 has simple and readily incorporated modifications, while Perrin's modifications are more synthetically challenging to access.

4. The term "UGUU/R motif" is unclear, especially in the Abstract where there is no underlining or other marking of cleavage site GU and thus no indication of the cleavage site (an alternative marking could be simply G|U). I finally figured out that "U/R" is meant to describe that the fourth of the four nucleotides can be either U or R, i.e., any of U, A, or G, based on the Figure 2 data. But I suggest to the authors that this "U/R" notation is unclear, especially when written as part of UGUU/R, and there is a better way to write this. Because the well-accepted IUBMB code for "any of U, A, or G but not C" is D, just like "either of A or G" is R, UGUU/R can instead be written as UGUD.

5. However, from the data in Figure 2B, there is not an obvious discontinuity in the experimental data to support the claimed UGUU/R (i.e., UGUD) motif. For normalized v0, UGUA is about 40% worse than UGUU, and in turn, UGUC is about 40% worse than UGUA. Why is the functional line drawn between UGUA and UGUC? It would be simpler, and consistent with the data, to say that the motif is UG|UN, i.e., just UG|U. This is also consistent with the activity data in Figure 2C, even the first data set for GATA3.

6. In Figure 4A, it is unclear why the secondary structures of the G12V (left) and WT (right) RNA sequences are shown with completely different folds. As stated in the figure caption, the sole difference between the two RNA sequences is a single point mutation, G (WT) to U (G12V) at position 29 in the 60 nt sequence. Position 29 is shown as unpaired in both secondary structures: it is not part of any structural element in G12V, and it is part of a single-stranded triloop in WT. Either G or U can be accommodated in either fold. Also, the folds that are shown do not seem to be the folds predicted using the standard algorithm. A quick use of the UNAFold web server with both RNA sequences gave neither fold for either sequence. Unlike both folds in Figure 4A, UNAFold shows the 5'-segment as entirely unpaired in all low-energy folds. (Minor: WT sequence, U51

accidentally not circled.)

Reviewer #3:

Remarks to the Author:

The authors report the chemical evolution of the classic Dz 10-23 framework optimized through a series of iterative design, build, test, and learn cycles intended to uncover permissive sites where chemical modifications could facilitate catalytic activity under physiological conditions. This addresses an immensely important and unsolved problem in the field, as currently the use of therapeutic DNAzymes have been hindered by limited activity outside of non-physiological conditions of very high Mg(II) concentrations and pH.

The authors take a novel, multifaceted approach, including introducing chemical modifications of the sugar moieties (XNAs), modifications to the binding arms, and strategic positioning of 2 phosphorothioate linkages. The results report a new enzyme highly modified 10-23 DNAzyme achieves ~65 turnovers in 30 minutes under buffers meant to simulate near-physiological conditions, a feat rivaled only by the unmodified parent sequence under forcing conditions of elevated Mg²⁺ and pH. Selectivity of preferred cleavage motifs was validated against a diverse set of known and novel DNAzyme targets. Biochemical assays support an autonomous RNA cleavage mechanism independent of RNase H1 engagement, and the designed enzyme was shown to exhibit persistent allele-specific knock-down of an endogenous mRNA encoding an undruggable oncogenic KRAS target.

The potential impact of these results is immense! This work makes a breakthrough that is of broad interest and that will considerably advance the field. The manuscript is well-motivated, concise, and clearly written.

One aspect where the discussion falls short is in the description of RNA cleavage mechanism, and primary focus on recent NMR and molecular simulation work that, while valuable, may not be representative of an active state. The authors should bring into the discussion key papers that outline RNA and DNA catalyzed RNA-cleavage mechanisms and may shed light more powerful potential design principles.

Specific concerns that should be addressed:

1. The authors comment: "Recent structural insights into the folding and dynamics of Dz 10-23 (Fig. 1A, Supplementary Fig. 1) provide a valuable starting point for designing next-generation DNAzymes that can function with improved stability and catalytic activity." The structural insights mentioned were from NMR-averaged precatalytic structure of 10-23 bound to an RNA substrate and MD simulations where the "inline fitness" of the nucleophile was very poor. This suggests the NMR structures, and likely the MD simulation ensembles do not represent an active state competent to carry out the chemical steps of RNA cleavage, and hence may be limited in their ability to provide guidance for design. Several RNA-cleaving ribozymes and at least one DNAzyme that utilize a G nucleobase to activate the nucleophile of the RNA substrate share a common active site structural motif known as the L-platform/L-scaffold [RNA (2020) 26, 111: 10.1261/rna.071894.119] that may provide deeper insight for design, in addition to comparative analysis of active sites of RNA-cleaving enzymes [ACS Catal. (2018) 8, 314: 10.1021/acscatal.7b02976]. This should be discussed.

2. It has been pointed out by Breaker that discussions of RNA-cleavage mechanisms in the literature can be enhanced by using common language to discuss different catalytic strategies. In this work, the authors focus on what is referred to as inline fitness (alpha catalysis) and nucleophile activation by a general base (gamma catalysis). The authors might wish to look closely at the recent review of catalytic strategies of RNA-cleaving nucleic acid enzymes [ACS Chem. Biol. (2019) 14, 1068: 10.1261/rna.071894.119].

Reviewer #1 (Remarks to the Author):

The manuscript by Nguyen et al. presents the systematic development of new DNAzyme (Dz) variants using a comprehensive set of chemical modifications. The approach converges on a greatly improved new variant (Dz 46) that shows significantly more activity under near physiological conditions. In addition, specific features of Dz 46 are presented, including an extended cleavage site preference, activity in the presence of RNase H1, and allele specificity in cell-free and cell-based KRAS model systems. Overall, the manuscript is of very high scientific quality and presents very important new results that will have a strong impact in the field of RNA-cleaving XNAs and could further advance the development of Dz-based therapeutics. I have only a few comments that should be clarified to further improve the manuscript.

Moderate Points:

1. Fig. S15A: In general, the ATP competition test is very interesting, however, the margin of error in the data presented seems to be quite large, which (partially) prevents a clear conclusion. If possible, the data and its interpretation would benefit from more data points (replicates) to reduce the margin of error. This could help to strengthen this important aspect.

Response: These experiments were redone in triplicate. The new data appears in Figure S16A with higher reproducibility.

2. Fig. 2: Since the effects of the cleavage site neighbours differ from previous results, it would be interesting to include comparative data on the Dz 1 variant to evaluate whether this is caused by the chemical modifications or is "just" selective for the sequences used.

Response: The manuscript was revised to contain comparative data on the nearest neighbor analysis for Dz 1 variants (see Fig. S17). Reactions were performed under conditions of 1 mM MgCl₂ and 5 mM MgCl₂. Reactions performed under elevated Mg²⁺ concentrations are more similar to the trend observed for Dz 46, suggesting that chemical modifications favor a conformation that is more similar to the unmodified DNAzyme under elevated Mg²⁺ conditions.

3. Fig. S16 B: Since a longer RNA target has lower activity (as expected), it would be interesting as well to see how Dz 1 (and possibly Dz 4) compares.

Response: The manuscript was revised to contain new data evaluating Dz 1 variants against the longer RNA substrate (see Fig. S19). As expected, Dz 1 functions with reduced activity as compared to Dz 46.

4. Fig. 2C: While it is interesting that the cleavage-site site preference is transferable to other systems, it would be equally interesting to see if the increased activity is also transferred to these systems. Therefore, please include the absolute values of the velocities already recorded (e.g., using a second y-axis). In addition, it would be interesting to see if the improvements of Dz 46 are also generally transferable to these systems, e.g., by comparison with the respective Dz 1 system (e.g., fold change in relative activity).

Response: The data provided in Figure 2C were updated to include absolute values for velocity along a second y-axis. Additionally, new data was added to the supplementary material showing that the enhanced activity observed for Dz 46 relative to Dz 1 is transferrable to other targets (Figure S13).

5. Fig. 3: The data shown in Fig. 3 are "good to have," but they do not add too much value to the manuscript. In this regard, the data in Fig. 4C are much more relevant. Also, the discussion of these data

can be considerably shortened, e.g., autonomous cleavage has also been shown in several previous assays using a longer RNA substrate. I would therefore recommend moving Fig. 3 to the SI. Instead, I would recommend moving some data of the most central modifications from the SI to the main text.

Response: The extent to which DNazymes mediate gene silencing in cells via intrinsic catalytic activity versus RNase H-mediated antisense mechanisms is currently a topic of intense debate (Refs. 21, 29, 30). To address this issue, we quantified the level of human RNase H1 levels in three different cell lines (to our knowledge, this is the first report of RNase H1 quantification), and then used this information to demonstrate that Dz 46 catalyzes RNA cleavage independently of RNase H at physiologically relevant concentrations. For these reasons, we feel that the data presented in Fig. 3 are central to our argument and should remain in the main text.

6. In the RNase experiments it would, however, be good to also have data with a ratio of 10:1 (RNA:Dz), as the different turnover conditions might affect RNase differently (as already potentially seen in Fig. 4C).

Response: The supplementary material was updated to include new data showing the RNase H experiment performed under multiple turnover conditions. The results observed under stoichiometric (Fig. 4C) and multiple-turnover conditions (Fig. S20) are similar.

7. Fig. 5C: It would be helpful to also plot/discuss absolute values to e.g., see changes in wt levels (it appears that this is affected at 24 h and would be relevant for therapeutic applications).

Response: The supplementary document was updated to contain plots of the absolute values for the data shown in Figure 5C (see Fig. S21A). The new figure also contains agarose gel images of the RFLPA analysis and GAPDH loading control.

8. It should be briefly discussed that the stabilization of the binding arms due to the chemical modifications alters the dissociation temperatures of the respective Dz:RNA complexes (pre- and post-catalytic), thus favoring in particular the multiple turnover conditions of the new variants over the Dz 1 variant and all other variants without modifications in the arm regions, especially Dz 4 (when using assay temperatures that are not ideal for Dz 1/4 turnover). The authors should either briefly discuss this aspect and/or test the Dz 1/4 variant at different temperatures to find their optimal turnover condition.

Response: The main text states that we focused our designs around an 8+7 binding arm configuration to provide a balanced solution to the problem of how to enhance enzyme kinetics while avoiding product inhibition caused by increased thermodynamics of substrate binding. We also state that all of the reactions were performed at the physiologically relevant temperature of 37°C, which is ultimately needed for future drug discovery efforts. Additionally, we modified the text to include a sentence acknowledging our efforts to modify the binding arms for optimal RNA binding affinity.

Minor points:

9. Fig. S4: In the main text Fig. S4 should be explained/mentioned before the results of Fig. S5 are discussed.

Response: The text was modified to include a description of Fig. S4 before Fig. S5.

10. Fig. S5: Unlike most other plots, it appears that these data do not allow a firm conclusion as to whether the observed slightly increased/decreased activities are truly significant. However, this set of experiments does a good job of identifying modifications that render the Dz inactive and modifications that keep it

active (with roughly comparable activities). Since the subsequent figures break down the respective modifications, I would recommend interpreting the data of Fig. S5 in the text as yes/no only.

Response: The text was modified to permit a more general interpretation of Fig. S5.

11. The Usage of 'OMe' is not consistent in the supplement (e.g., Fig. S8A Table 'OMe', and several other positions). Please crosscheck and use consistent nomenclature.

Response: The text was modified so that the "OMe" nomenclature is used consistently.

12. Is there a reason why variant 44 was not evaluated further? It does not seem to differ from the three selected variants.

Response: During the course of our experiments, we consistently noticed that Dz 46 functions with faster catalytic activity than Dz 44 in multiple turnover experiments. However, we do note that Dz 44 is also very strong enzyme worthy of further evaluation.

13. It would be helpful to include a table of the key modifications/constructs in Fig. 1 (maybe remove gels in Fig. 1C to SI).

Response: The supplementary material (see Fig. S11) was modified to include a diagram of the key modifications found in the DNAzymes described in Figure 1.

14. The way the cleavage velocity is calculated (i.e. using 10-15% cleavage product), could potentially change between the assays using 10:1 and 100:1 (RNA:Dz) ratios. How does the velocity compare between the data shown in Fig. S12B,C vs. Fig. 1D,E ?

If this is an issue it should be briefly discussed and/or a more robust analysis used.

Response: Velocity measurements were calculated from multiple turnover reactions performed under the more stringent condition of 100:1 (S:E). This was necessary because the fastest variants yielded >15% cleavage within the first 30 seconds when the less stringent conditions were used, making such values unreliable as kinetic terms. This information is provided in the legend of Figure 1.

15. Please rewrite phrase '10-30 fold less dependent' (page 7) to e.g., 'strongly reduced dependency on Mg²⁺' (since the data do not quantify the 'dependency').

Response: This text was modified to correct this oversight.

16. The interpretation of improved cofactor (Mg²⁺) binding due to the modifications is speculative. The modifications could also render cofactor binding obsolete at some of the binding sites. Please rewrite the respective sentences accordingly.

Response: The text was revised for clarity.

17. The authors highlight the additional constrains of the cleavage site as an exclusively positive effect. While reduced number of target sites will indeed simplify the selection process, it will also impose additional constrains. Please rewrite the respective sentences accordingly.

Response: The text was updated to include this additional constraint, which could affect some mRNA targets.

18. While the new variants are certainly impressively effective, phrases like 'making it the fastest' are still difficult to assess, since most previous Dz variants have been investigated under different conditions. In addition, regular 10-23 Dz (Dz 1) variants with considerably higher activity as compared to the used Dz 1 (KRAS) variants have been reported, making the comparison even more difficult. Therefore, please use a less 'absolute' wording.

Response: The text was revised for generality by using less absolute wording.

19. I would recommend not selecting single parameters, such as ">250 times more effective than" when making quantitative comparisons with results from other groups; this oversimplifies "efficiency." In this example, the discussed variants indeed show significantly reduced multiple turnover rates when using minimal RNA substrates; however, they do not seem to suffer as much when using longer RNA constructs and also show comparable effects to Dz 46 in cellular assays. Therefore, please either include a more detailed discussion or write the relevant statement more generally.

Response: The sentence was revised for generality by removing the quantitative metrics.

20. While the used buffer is well selected and reflects ion concentrations normally found under physiological conditions, many other factors will affect the in vivo activity of Dz. The authors already demonstrate that 1 mM ATP (which is still below frequently found cellular concentrations) will reduce activity by 50% and many other molecules may also interfere with Mg²⁺ and/or the DNA/RNAs. Therefore, please rephrase all sentences referring to in vitro conditions from "under physiological conditions" to "under near physiological conditions"

Response: The requested changes have been made to the text.

Reviewer #2 (Remarks to the Author):

The authors describe the strategic chemical evolution of the RNA-cleaving 10-23 DNAzyme for improved catalytic activity under physiological conditions. As the authors describe, achieving high-activity DNAzyme function in vivo has been a long-standing challenge, due to the generally high divalent metal ion (Mg²⁺) requirement of DNAzymes as well as the alternative RNase H-based antisense mechanism enabled by the Watson-Crick DNAzyme binding arms. Here they use the high-resolution structural information published by others in ref. 21 (Nature 2022) to design a stepwise series of chemical changes in 10-23, eventually finding DNAzyme 46 (Dz 46). They show that Dz 46 has the highest reported multiple turnover of any DNAzyme for RNA cleavage under physiological conditions. The data appear strong to support an autonomous, RNase H-independent cleavage mechanism. Allele-specific RNA cleavage is established, including in a cellular assay by PCR-RFLP analysis. I support publication in Nature Communications after suitable revisions to address the following comments.

1. The claim of "perfect specificity" (page 10) of the DNAzyme for the two KRAS mRNA targets can surely be quantified more precisely, based on the experimental upper limit of "no cross-reactivity was observed for either DNAzyme against the opposing substrate". For instance, if the detection limit in Figure 4B is <1%, then the specificity would be >99%. Any such value should be corrected for the % yield in the relevant

active lane (e.g., <1% versus 50% yields would mean >98% specificity).

Response: The text was revised to read ">99%."

2. "We suggest that the approach taken was sufficiently general that it could be applied to other DNAzymes that have been developed..." (page 12). This sounds right, but only as long as those other DNAzymes also have the detailed structural information analogous to that for 10-23 in ref. 21 - which I understand took those authors about a decade to obtain! I think this warrants a disclaimer in the main text that application to other DNAzymes will require structural information that is not trivial to obtain.

Response: The text was revised to note the importance of structural information.

3. It is surprising not to see any direct citation of the extensive work of David Perrin (U. British Columbia) on seeking metal-independent RNA-cleaving DNAzymes. There is a sort of hint of this in the citation of ref. 12, which is a review from the authors themselves rather than Perrin. The authors can correctly note that Dz 46 has simple and readily incorporated modifications, while Perrin's modifications are more synthetically challenging to access.

Response: The text was updated to include the suggested citation.

4. The term "UGUU/R motif" is unclear, especially in the Abstract where there is no underlining or other marking of cleavage site GU and thus no indication of the cleavage site (an alternative marking could be simply G|U). I finally figured out that "U/R" is meant to describe that the fourth of the four nucleotides can be either U or R, i.e., any of U, A, or G, based on the Figure 2 data. But I suggest to the authors that this "U/R" notation is unclear, especially when written as part of UGUU/R, and there is a better way to write this. Because the well-accepted IUBMB code for "any of U, A, or G but not C" is D, just like "either of A or G" is R, UGUU/R can instead be written as UGUD.

Response: The text was revised to include "UGUD" as the preferred motif. D was defined as the IUPAC nomenclature for nucleotide abbreviations.

5. However, from the data in Figure 2B, there is not an obvious discontinuity in the experimental data to support the claimed UGUU/R (i.e., UGUD) motif. For normalized v_0 , UGUA is about 40% worse than UGUU, and in turn, UGUC is about 40% worse than UGUA. Why is the functional line drawn between UGUA and UGUC? It would be simpler, and consistent with the data, to say that the motif is UG|UN, i.e., just UG|U. This is also consistent with the activity data in Figure 2C, even the first data set for GATA3.

Response: Although we chose UGUD to represent the preferred cleavage motif Dz 46, where D is the IUPAC nomenclature for A, G, and T/U, the UGUU motif consistently outperforms the other motifs. Nevertheless, we chose to include the UGUG and UGUA motifs in the preferred set so that practitioners could have other options to consider when the preferred UGUU is either missing, occluded, or lies in a suboptimal sequence context. In our experience, UGUC is a suboptimal motif that should be avoided when better options are available.

6. In Figure 4A, it is unclear why the secondary structures of the G12V (left) and WT (right) RNA sequences are shown with completely different folds. As stated in the figure caption, the sole difference between the two RNA sequences is a single point mutation, G (WT) to U (G12V) at position 29 in the 60 nt sequence. Position 29 is shown as unpaired in both secondary structures: it is not part of any structural element in G12V, and it is part of a single-stranded triloop in WT. Either G or U can be accommodated in either fold.

Also, the folds that are shown do not seem to be the folds predicted using the standard algorithm. A quick use of the UNAFold web server with both RNA sequences gave neither fold for either sequence. Unlike both folds in Figure 4A, UNAFold shows the 5'-segment as entirely unpaired in all low-energy folds. (Minor: WT sequence, U51 accidentally not circled.)

Response: We re-analyzed the WT and G12V sequences using UNAFold version 2.3 and obtained the same results. However, we do note that our rendering of the structures differs from the UNAFold prediction by a flattening of the single-stranded bulges, which was done for illustrative purposes. However, we did revise Figure 4A to include a circle around residue U51.

Reviewer #3 (Remarks to the Author):

The authors report the chemical evolution of the classic Dz 10-23 framework optimized through a series of iterative design, build, test, and learn cycles intended to uncover permissive sites where chemical modifications could facilitate catalytic activity under physiological conditions. This addresses an immensely important and unsolved problem in the field, as currently the use of therapeutic DNAzymes have been hindered by limited activity outside of non-physiological conditions of very high Mg(II) concentrations and pH.

The authors take a novel, multifaceted approach, including introducing chemical modifications of the sugar moieties (XNAs), modifications to the binding arms, and strategic positioning of 2 phosphorothioate linkages. The results report a new enzyme highly modified 10-23 DNAzyme achieves ~65 turnovers in 30 minutes under buffers meant to simulate near-physiological conditions, a feat rivaled only by the unmodified parent sequence under forcing conditions of elevated Mg²⁺ and pH. Selectivity of preferred cleavage motifs was validated against a diverse set of known and novel DNAzyme targets. Biochemical assays support an autonomous RNA cleavage mechanism independent of RNase H1 engagement, and the designed enzyme was shown to exhibit persistent allele-specific knock-down of an endogenous mRNA encoding an undruggable oncogenic KRAS target.

The potential impact of these results is immense! This work makes a breakthrough that is of broad interest and that will considerably advance the field. The manuscript is well-motivated, concise, and clearly written.

One aspect where the discussion falls short is in the description of RNA cleavage mechanism, and primary focus on recent NMR and molecular simulation work that, while valuable, may not be representative of an active state. The authors should bring into the discussion key papers that outline RNA and DNA catalyzed RNA-cleavage mechanisms and may shed light more powerful potential design principles.

Specific concerns that should be addressed:

1. The authors comment: "Recent structural insights into the folding and dynamics of Dz 10-23 (Fig. 1A, Supplementary Fig. 1) provide a valuable starting point for designing next-generation DNAzymes that can function with improved stability and catalytic activity." The structural insights mentioned were from NMR-averaged precatalytic structure of 10-23 bound to an RNA substrate and MD simulations where the "inline fitness" of the nucleophile was very poor. This suggests the NMR structures, and likely the MD simulation ensembles do not represent an active state competent to carry out the chemical steps of RNA cleavage, and hence may be limited in their ability to provide guidance for design. Several RNA-cleaving ribozymes and at least one DNAzyme that utilize a G nucleobase to activate the nucleophile of the RNA substrate share a common active site structural motif known as the L-platform/L-scaffold [RNA (2020) 26, 111: 10.1261/rna.071894.119] that may provide deeper insight for design, in addition to comparative analysis of

active sites of RNA-cleaving enzymes [ACS Catal. (2018) 8, 314: 10.1021/acscatal.7b02976]. This should be discussed.

Response: The text was modified to include a discussion (with citations) of how the L-platform could shed light on future Dz design principles.

2. It has been pointed out by Breaker that discussions of RNA-cleavage mechanisms in the literature can be enhanced by using common language to discuss different catalytic strategies. In this work, the authors focus on what is referred to as inline fitness (alpha catalysis) and nucleophile activation by a general base (gamma catalysis). The authors might wish to look closely at the recent review of catalytic strategies of RNA-cleaving nucleic acid enzymes [ACS. Chem. Biol. (2019) 14, 1068: 10.1261/rna.071894.119].

Response: The text was modified to include a discussion (with citations) of how RNA cleavage mechanisms described by Breaker play an important role in DNAzyme catalysis.

Reviewers' Comments:

Reviewer #1:

Remarks to the Author:

The authors have done an excellent job in revising the manuscript. I would like to congratulate them on their exciting work and thank them for taking the extra effort to adequately address the rather long list of suggestions, which I believe has further improved the manuscript.

Reviewer #2:

Remarks to the Author:

The authors have revised their manuscript in response to the three previous reviewers. I was Reviewer 2. Everything now appears fine and acceptable for publication, after two minor adjustments mentioned below.

1. The degenerate nucleotide codes such as R and D were established by IUBMB, not IUPAC (page 4, line 87). See <http://dx.doi.org/10.1093/nar/13.9.3021> and <https://www.jstor.org/stable/26785>. IUBMB was called the International Union of Biochemistry at that time.

2. Page 13, line 330, it is unclear why all three of GUD are underlined, versus only GU to mark the cleavage site as is done elsewhere throughout manuscript.

Reviewer #3:

Remarks to the Author:

The authors have been very thorough in addressing all the constructive criticisms of the previous review, and the manuscript has been substantially improved in clarity and scientific content as a result. No further revisions are recommended.

Reviewer #1 (Remarks to the Author):

The authors have done an excellent job in revising the manuscript. I would like to congratulate them on their exciting work and thank them for taking the extra effort to adequately address the rather long list of suggestions, which I believe has further improved the manuscript.

Response: We appreciate the reviewer's positive assessment of the revised manuscript and thank them for their constructive comments.

Reviewer #2 (Remarks to the Author):

The authors have revised their manuscript in response to the three previous reviewers. I was Reviewer 2. Everything now appears fine and acceptable for publication, after two minor adjustments mentioned below.

1. The degenerate nucleotide codes such as R and D were established by IUBMB, not IUPAC (page 4, line 87). See <http://dx.doi.org/10.1093/nar/13.9.3021> and <https://www.jstor.org/stable/26785>. IUBMB was called the International Union of Biochemistry at that time.

Response: The text was updated by replacing IUPAC with IUBMB, and by including the suggested citation for IUBMB nomenclature.

2. Page 13, line 330, it is unclear why all three of GUD are underlined, versus only GU to mark the cleavage site as is done elsewhere throughout manuscript.

Response: The text was updated by removing the underline format for D.

Reviewer #3 (Remarks to the Author):

The authors have been very thorough in addressing all the constructive criticisms of the previous review, and the manuscript has been substantially improved in clarity and scientific content as a result. No further revisions are recommended.

Response: We appreciate the reviewer's positive assessment of the revised manuscript and thank them for their constructive comments.